# Lactate Activates Germline and Cleavage Embryo Genes in Mouse Embryonic Stem Cells

**DOI:** 10.3390/cells11030548

**Published:** 2022-02-04

**Authors:** Qing Tian, Li-quan Zhou

**Affiliations:** Institute of Reproductive Health, Tongji Medical College, Huazhong University of Science and Technology, Wuhan 430030, China; d202081624@hust.edu.cn

**Keywords:** lactate, histone lactylation, cell fate, germline gene, zygotic genome activation

## Abstract

Lactate was recently found to mediate histone lysine lactylation and facilitate polarization of M1 macrophages, indicating its role in metabolic regulation of gene expression. During somatic cell reprogramming, lactate promotes histone lactylation of pluripotency genes and improves reprogramming efficiency. However, the function of lactate in cell fate control in embryonic stem cells (ESCs) remains elusive. In this study, we revealed that lactate supplementation activated germline genes in mouse ESCs. Lactate also induced global upregulation of cleavage embryo genes, such as members of the *Zscan4* gene family. Further exploration demonstrated that lactate stimulated H3K18 lactylation accumulation on germline and cleavage embryo genes, which in turn promoted transcriptional elongation. Our findings indicated that lactate supplementation expanded the transcriptional network in mouse ESCs.

## 1. Introduction

Embryonic stem cells (ESCs), derived from inner cell mass of blastocysts, possess an unlimited capacity for proliferation, self-renewal, and multipotent differentiation [1], offering biologists a valuable experimental system to uncover the regulatory mechanism of pluripotency maintenance and differentiation during development. Furthermore, in vitro cultured ESCs occasionally overcome epigenetic barriers and transiently reach totipotent status with two-cell-like transcriptome and chromatin features [2,3,4]. Hence, in vitro cultured ESCs are also widely used to explore cell fate transition from pluripotency to totipotency [5,6,7].

Cellular metabolism is the most fundamental biological process satisfying energy demands to keep cells alive, and discrepancies in metabolism patterns reflect distinct cell status [8,9]. Progressing beyond crucial roles in energy homeostasis, the last decade has witnessed significant advances in our understanding of metabolic regulation of epigenetic modifications, gene expression, and cell fate change [10,11,12]. Like most rapidly dividing cells, pluripotent stem cells sustain high glycolysis to feed both cellular demands for building blocks such as nucleotides, phospholipids, and amino acids and energy needs [10,13]. Still, discrepant metabolic profiles associate with distinct pluripotent states, and cells reprogram their metabolic pattern during cell fate transition [14,15,16]. Since the discovery of acetyl-CoA as the substrate of histone acetyltransferases (HATs) to regulate histone acetylation in mammalian cells [17,18], increasing numbers of metabolic intermediates have been found to play pivotal roles in epigenetic modification. S-adenosylmethionine (SAM), a key intermediate of the one-carbon cycle, is the methyl donor for histone and DNA methylation reactions [8,19]. Deprivation of SAM causes downregulation of H3K4me3 and differentiation in ESCs [20,21]. α-ketoglutarate (αKG), an intermediate of the TCA cycle, is a required cofactor for both histone and DNA demethylase [10,13]. In ESCs, αKG promotes DNA and histone demethylation, activates the pluripotent gene, and maintains the pluripotent state [22,23]. Although an increasing number of metabolic intermediates have been found to play a regulatory role in epigenetic regulation of cell fate, the causal relationship between metabolic reprogramming and cell fate shift requires further investigation.

Lactate is an end product of glycolysis that is reused for gluconeogenesis in the liver. In tumor cells, the lactate produced by glycolysis facilitates the formation of an acidic tumor microenvironment, which reinforces cancer invasion and suppresses antitumor immunity [24,25]. Recent studies have identified a novel function for lactate whereby it is utilized for histone lysine lactylation to promote the transition from inflammatory to reparative macrophages and activate homeostatic gene expression, maintaining immune homeostasis [26,27]. Discovery of lactoyl-CoA (lactyl-CoA) in mammalian cells indicated that lactoyl-CoA generated by glucose metabolism was a possible biochemical link between lactate and histone lactylation in vivo [24,28]. A cell-free, recombinant chromatin-templated histone modification and transcription assay demonstrated that lactoyl-CoA activated transcription in a p53-dependent, p300-mediated manner [27], and deprivation of p300 reduced histone lactylation both in vitro and in vivo [27,29], indicating p300 as a potential histone lactyltransferase [24,30]. Moreover, lactate was found to play roles in lung fibrosis, cell differentiation and reprogramming, and lactylation of nonhistone proteins [24]. As an important metabolic intermediate, the role of lactate and lactate-induced histone lactylation in pluripotency maintenance and differentiation is still unclear. In this study, we uncovered the function of lactate and lactate-induced histone lactylation in cell fate control in mouse ESCs.

## 2. Materials and Methods

### 2.1. Cell Culture

The mouse ESC line AB2.2 was cultured on mouse embryonic fibroblast (MEF) feeder cells with ES cell medium supplemented with 15% fetal bovine serum (Hyclone, Logan, UT, USA, cat. no. SH30396.03), 1000 U/mL leukemia inhibitory factor (LIF, Millipore, Burlington, MA, USA, cat. no. ESG1107), 3 μM CHIR99021 (LC Laboratories, Woburn, MA, USA, cat. no. C-6556), and 1 μM PD0325901 (LC Laboratories, cat. no. P-9688). Fifty millimolar lactate (Sangon Biotech, Shanghai, China, cat. no. A604046) was added and maintained for 24 h before experimental procedures. Cells were cultured in a humidified chamber at 37 °C with 5% CO_2_. For culture of mouse ESC lines, the medium was refreshed daily, and cells were routinely passaged every 2 days.

### 2.2. Western Blot

Cells were lysed on ice with gentle stirring for 10 min in 0.5 mL of TBE buffer containing 0.5% Triton X-100 (Biosharp, Anhui, China. cat. no. BS084) (*v*/*v*) and 2 mM phenylmethylsulfonyl fluoride (PMSF)(Solarbio, Beijing, China. cat. no. P0100). The lysates were centrifuged at 6500× *g* for 10 min at 4 °C to remove the supernatant. Retained nuclei were washed twice using TEB and centrifuged as before. The nuclei were then resuspended in 0.2 N HCl at 4 °C overnight. Samples were centrifuged at 6500× *g* for 15 min at 4 °C to remove the nuclei debris the next day. Samples were subjected to Western blot analysis with the appropriate antibodies (anti-H4K8la, cat. no. PTM-1415; anti-H4K12la, cat. no. PTM-1411RM; and anti-H3K18la, cat. no. PTM-1406RM. PTM BIO, Hangzhou, China) after neutralizing HCl with 2M NaOH at 1/5 of the volume of the supernatant.

### 2.3. RNA Isolation and Quantitative RT-PCR (qRT-PCR)

Total RNA was extracted using a TRI reagent (Sigma, St. Louis, MO, USA, cat. no. T9424) following the manufacturer’s procedure. The purity and concentration of RNA samples were determined with a Nanodrop ND-2000 spectrophotometer (Thermo Fisher Scientific, Waltham, MA, USA). A reverse-transcriptional reaction was performed with a Hifair III 1st Strand cDNA Synthesis Kit (Yeasen, Shanghai, China. cat. no. 11139ES60) according to the manufacturer’s procedure. qRT-PCR was performed with SYBR green master mix (Yeasen, cat. no. 11203ES08) on a StepOnePlus™ Real-Time PCR System with Tower (Applied Biosystems, Foster City, CA, USA) according to the manufacturer’s instructions. For primer sequences, see Appendix A.

### 2.4. ChIP-Seq

ChIP-seq was performed using a Hyperactive In-Situ ChIP Library Prep Kit for Illumina (pG-Tn5) (Vazyme, Nanjing, China, cat. no. TD901) according to the manufacturer’s procedure with the appropriate antibodies (anti-H3K18la, PTM-1406RM, and PTM BIO; anti-H3K4me3, 9727, and CST, Danvers, MA, USA; and anti-PolII, 39097, and active motif, Carlsbad, CA, USA).

### 2.5. RNA-Seq Dataset Analysis

Raw reads were processed with cutadapt v1.16 (https://cutadapt.readthedocs.io, accessed on 24 September 2021) to remove adapters and perform quality trimming with default parameters. Trimmed reads were mapped to the mouse genome (GENCODE release M23) using STAR (v2.5.1b) with default settings. Reads were counted in exons of the mouse genome (GENCODE release M23) using the STAR-quantMode GeneCounts setting. Differential expression of genes for all pairwise comparisons was assessed by DESeq2 v1.24.0 with internal normalization of reads to correct for library size and RNA composition bias. Differentially regulated genes in the DESeq2 analysis were defined as those that were more than two-fold increased or decreased with an adjusted *p*-value < 0.05. RSEM was used to calculate the FPKM value for each gene. Gene ontology was performed using Metascape (https://metascape.org, accessed on 12 October 2021) [31]. For GO term IDs, see Appendix A. The enrichment bubble dot and heatmap were plotted on a website (http://www.bioinformatics.com.cn, accessed on 12 October 2021).

### 2.6. ChIP-Seq Dataset Analysis

Raw reads were processed with cutadapt v1.8.1 to remove adapters and perform quality trimming. Trimmed reads were mapped to the UCSC mm10 assembly using Bowtie2 with default parameters. Deeptools was used for normalization to draw a read density plot and heatmap from bigwig files for visualization of ChIP-seq data.

### 2.7. Microarray Dataset Analysis

Microarray datasets (GSE45181) were analyzed using GEO2R, an interactive web tool. Datasets GSM1098610, GSM1098612, and GSM1098614 were used as a control group, and datasets GSM1098611, GSM1098613, and GSM1098615 were used as a treatment group (*Max* knockdown).

### 2.8. Statistical Analysis

The two-tailed Student’s t-test and Wilcoxon rank sum test were used to calculate *p* values. Statistically significant values for *p* < 0.05, *p* < 0.01, and *p* < 0.001 are indicated by single, double, and triple asterisks, respectively.

## 3. Results 

### 3.1. Lactate Supplementation Stimulated H3K18 Lactylation in Mouse ESCs

It was reported that exogenous lactate stimulates lactylation of histone lysine residues in cells [27]. Hence, we added 50 mM lactate to ES culture media to enhance histone lactylation. As expected, lactate supplementation significantly improved lactylation of H3K18 (H3K18la) after 24 h treatment. We found that H4K8la and H4K18la modifications were slightly weakened upon lactate supplementation, probably due to crosstalk among different histone lactylation modifications (Figure 1A). To reveal the effect of lactate supplementation on cell pluripotency, we examined the expression of pluripotent genes. The expression of pluripotent genes, such as *Oct4*, *Nanog,* and *Sox2*, was not affected upon lactate supplementation (Figure 1B). These results indicated that lactate supplementation strengthened H3K18la without affecting pluripotent cell identity of mouse ESCs. We further performed a ChIP-seq assay to illustrate the distribution of H3K18la in mouse ESCs, and examined the H3K18la signals at gene loci with different expression levels (only genes with FPKM >1 were analyzed) by RNA-seq. Interestingly, we noticed that genes expressed at higher levels had more a enriched H3K18la signal along the whole gene bodies compared to genes expressed at lower levels (Figure 1C). This result supported a positive correlation between H3K18la intensity and transcript abundance and indicated that many genes may be upregulated by lactate supplementation.

### 3.2. Lactate Supplementation Activated Germline Genes in mESCs

To study the potential influence of lactate supplementation on cell identity, we performed RNA-seq to uncover changes in the transcriptome. In general, there were 160 upregulated genes (|log_2_ fold-change| > 1, adjusted *p*-value < 0.05) and 145 downregulated genes (|log_2_ fold-change| < 1, adjusted *p*-value < 0.05) upon lactate supplementation (Figure 2A,B). Gene ontology analysis identified that there was enrichment in the downregulated genes of the response to virus, the CAMKK–AMPK signaling cascade, and the regulation of nervous system development. Remarkably, we noticed significant enrichment of the germline gene population in upregulated genes indicated by GO terms including ‘meiotic DNA double-strand break formation’ and ‘DNA methylation involved in gamete generation’ (Figure 2C). Correspondingly, the heatmap of RNA-seq data showed upregulation of a bulk of typical germline genes such as *Dazl*, *Ddx4*, *Mael*, *Dmrt1*, and *Taf7l* (Figure 2D). Upregulation of germline genes was also confirmed by qRT-PCR (Figure 2E). 

### 3.3. Lactate Supplementation Induced Cleavage Embryo Genes in Mouse ESCs

In addition to germline genes, we also observed a significant upregulation of *Zscan4* family genes (Figure 3A), a set of zygotic genome activation (ZGA) marker genes expressed specifically in two-cell mouse embryos and six to eight cell human embryos [32,33]. To further uncover the global expression change of ZGA genes, we reanalyzed the RNA-seq dataset to obtain upregulated genes in MERVL^+^Zscan4^+^ ESCs relative to MERVL^–^Zscan4^–^ ESCs (|log_2_ fold-change| > 1, adjusted *p*-value < 0.05) for further analysis (Appendix A) [34]. A globally increased abundance of upregulated genes in MERVL^+^Zscan4^+^ ESCs was also observed in lactate-supplemented ESCs (Figure 3B,C). Upregulation of ZGA genes, such as *Zscan4*, *Zfp352*, *Tcstv3*, and *Sp110*, was verified by qRT-PCR (Figure 3D). However, the expression of *Dux* was not changed (Figure 3E), a key regulatory factor that activates hundreds of ZGA genes and retroviral elements [35]. These results correspond to a previous discovery that lactate raised the ratio of two-cell-like ESCs [14].

### 3.4. Sequential Activation of Germline and ZGA Genes in Mouse ESCs 

To study the correlation between the expression of germline and ZGA genes, we analyzed the public transcriptome data in which ZGA and germline genes were activated. Zscan4c was reported to be a key activator of cleavage embryo genes in ESCs, so we analyzed RNA-seq data of mouse ESCs in which *Zscan4c* was overexpressed [36]. Expression of exogenous *Zscan4c* was almost 16 times higher than that of endogenous *Zscan4c* (Figure 4A). Activation of ZGA genes, such as *Zscan4* family genes, *Zfp352,* and *Dux*, was observed after *Zscan4c* overexpression (Figure 4B), while activation of germline genes was not observed (Figure 4C). For transcriptional regulation of germ-cell-specific genes in mouse ESCs, *Max* was reported to be the key repressor by RNA interference screen, whose knockdown in ESCs resulted in a global upregulation of germ-cell-specific genes [37]. Therefore, we analyzed the microarray data in which *Max* was knocked down to activate germline genes in mouse ESCs [37]. Successful suppression of *Max* was confirmed (Figure 4D), and significant upregulation of germline genes, such as *Ddx4*, *Dazl,* and *Mov10l1*, was also observed upon *Max* knockdown (Figure 4E). In addition, we found a global upregulation of ZGA genes, such as *Zscan4* family genes, *Tcstv1*, and *Zfp352* (Figure 4F). These results suggested that germline gene activation stimulated ZGA genes in ESCs, while ZGA gene activation had little impact on germline gene expression.

### 3.5. Lactate Supplementation Facilitated Transcriptional Elongation through Enhanced Histone Lactylation

To gain further mechanistic insights into regulation of germline and ZGA genes, we deciphered the occupancy landscape of H3K18la, H3K4me3, and RNA polymerase II (PolII) upon lactate supplementation. We found that there was remarkably increased deposition of H3K18la at the transcription start site (TSS), as well as a slightly increased accumulation at the gene body of upregulated genes (Figure 5A). Consistently, the intensity of H3K4me3, a marker of an active promoter, also increased at the TSS of upregulated genes (Figure 5B). Although no discrepant PolII localization around the TSS of upregulated genes was observed after lactate supplementation, increased PolII occupancy at the gene body was detected (Figure 5C). Our results indicated that lactate supplementation reinforced histone lactylation at the promoter and gene body and facilitated transcriptional elongation of target genes.

## 4. Discussion

In this study, we identified that lactate supplementation facilitated the expression of germline and ZGA genes through transcriptional elongation orchestration. In addition to recruitment and assembly of preinitiation complex (PIC), promoter-proximal pausing and release are other key steps in transcription regulation by PolII. While negative elongation factor (NELF) and DRB-sensitivity-inducing factor (DSIF) stabilize the paused Pol II, positive transcription elongation factor-b (P-TEFb) drives release of paused Pol II to begin productive elongation [38,39]. ELL is a component of the super elongation complex (SEC), which interacts directly with P-TEFb to promote elongation. p300 acetylates ELL and increases its stability [40]. Interestingly, p300 also acetylates both promoter-paused polymerase and gene-body-occupied polymerase, promoting PolII release and elongation efficiency [41]. Moreover, acetylation of promoter histone H3K18/K27 mediated by p300 stimulates the release of paused PolII and the recruitment of the super elongation complex (SEC), promoting productive elongation [42]. In this study, we observed accumulated PolII among the gene body but not on the promoter upon lactate supplementation. We suggest that H3K18la recruited transcription coactivators, possibly p300, and promoted transcription elongation. Additionally, we suggest that histone lactylation might facilitate PolII elongation by loosening nucleosome–DNA interaction and forming a platform for coactivators binding. There is a high distribution similarity between histone acetylation and lactylation among the genome [27]. Possible crosstalk between acetylation and lactylation warrants further study.

Eight to twelve percent of couples are infertile worldwide [43], and gametogenesis failure is one of the main causes. Therefore, establishment of an in vitro system for efficient generation of functional gametes is a promising approach for infertility treatment. Stem cells (SCs) including ESCs and iPSCs, possessing the ability to differentiate into all cell types, are valuable cell origins for in vitro differentiation toward germ cells. Though several methods and techniques have been identified to induce SCs’ differentiation into early germ cells, oocyte-like cells (OLCs), and male gametes [44,45,46,47], the efficiency is quite low. In our study, we discovered that lactate supplementation activates the expression of germline genes involved in DNA methylation and meiotic division in mESCs. We also found upregulation of a wealth of genes regulating piRNA metabolism, such as *Piwil2*, *Mael*, and *Mov10l1*. These results suggested that lactate supplementation facilitates differentiation of mouse ESCs toward germ cells. Whether and how lactate improves differentiation efficiency toward gametes warrants further study.

In ESCs, transient *Zscan4* expression upregulated homologous recombination genes and facilitated telomere elongation by homologous recombination [48], improving the developmental potency of ES cells and restoring the differentiation potential of long-term cultured ES cells [3]. Knockdown of *Zscan4* led to telomere shortening, genomic instability, and aneuploidy [48]. In addition, histone hyperacetylation and DNA demethylation were observed during transient *Zscan4* expression [49]. Mechanism studies indicated that ZSCAN4 recruited TET2 through its SCAN domain to target locus and promoted DNA demethylation [50] while promoting degradation of UHRF1 and DNMT1 to suppress DNA methylation [51]. In this study, we observed a significant upregulation of the *Zscan4* gene family and a global upregulation of ZGA genes upon lactate supplementation, and this may trigger other downstream events. Lactate is one of the major energy sources of cleavage embryos and promotes embryonic development to the morula stage [12,52]. Therefore, lactate may also regulate preimplantation development by activating ZGA and promoting telomere elongation and global DNA demethylation in early embryos.

## 5. Conclusions

In summary, we discovered for the first time that lactate supplementation activates the expression of germline and ZGA genes in mouse ESCs, especially the expression of germline genes and members of the *Zscan4* gene family. We propose that lactate stimulates H3K18la accumulation at germline and ZGA genes, which recruit cofactors to facilitate transcriptional elongation. 

## Figures and Tables

**Figure 1 cells-11-00548-f001:**
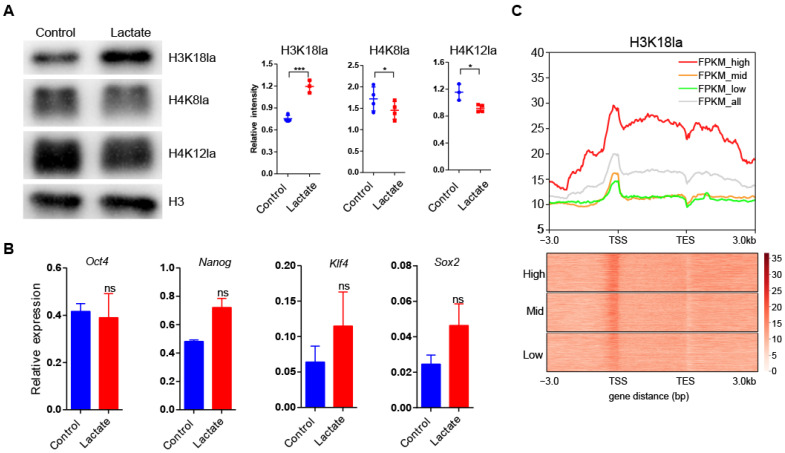
Lactate supplementation enhanced lactylation of H3K18 in mouse ESCs. (**A**) Western blotting of H3K18la, H4K8la, and H4K18la (**left**) and corresponding statistical analysis (**right**). Data are presented as means ± SD (*n* = 3, 4); *** *p* < 0.001; * *p* < 0.05; ns, *p*  >  0.05. Student’s t-test was used to calculate *p*-values. Histone H3 was used as the internal control. (**B**) qRT-PCR analysis of pluripotent genes in control and lactate-supplemented mouse ESCs. Data are presented as means ± SD (*n* = 3); ns, *p* > 0.05. Student’s *t*-test was used to calculate *p*-values. *Actb* gene was used as the internal control. (**C**) Average H3K18la enrichment along gene bodies and 3 kilobases (kb) upstream/downstream of gene bodies of the first third (**high**), the second third (**mid**), the posterior third (**low**), and all genes in WT ESC, sorted by FPKM (**upper panel**). Only genes with FPKM >1 were analyzed. Corresponding heatmap showing H3K18la enrichment along gene bodies and 3 kilobases (kb) upstream/downstream of gene bodies (**lower panel**).

**Figure 2 cells-11-00548-f002:**
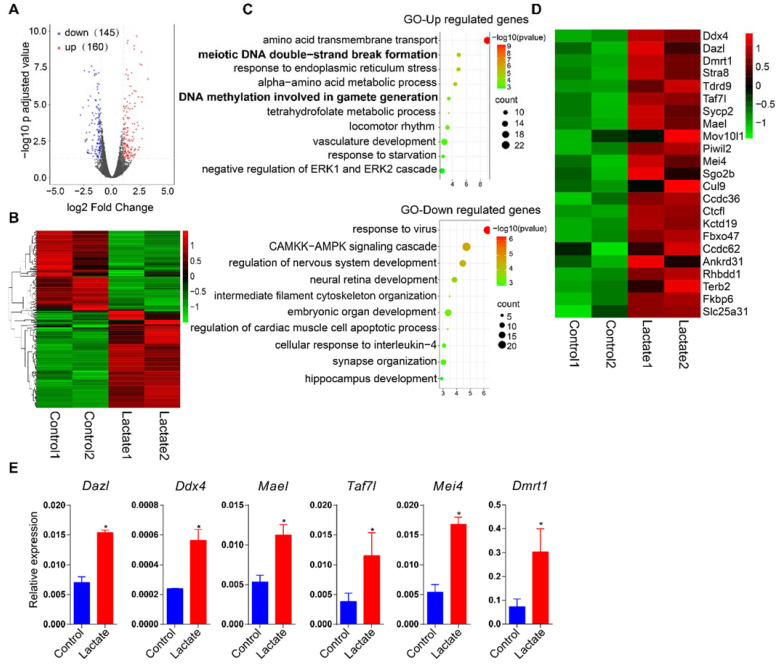
Lactate supplementation activated germline genes in mouse ESCs. (**A**) Scatter plot comparing transcripts between control and lactate-supplemented mouse ESCs. Gene expression changes with |log_2_ fold-change| > 1 in lactate supplemented mouse ESCs are highlighted with red (upregulation) or blue (downregulation), respectively. (**B**) Heatmap of misregulated genes upon lactate supplementation. (**C**) Gene ontology analysis of enriched terms of up- and downregulated genes. (**D**) Heatmap of representative upregulated germline genes. (**E**) qRT-PCR analysis of germline genes in control and lactate-supplemented mouse ESCs. Data are presented as means ± SD (*n* = 3), * *p* < 0.05. Student’s *t*-test was used to calculate *p*-values. *Actb* was used as the internal control.

**Figure 3 cells-11-00548-f003:**
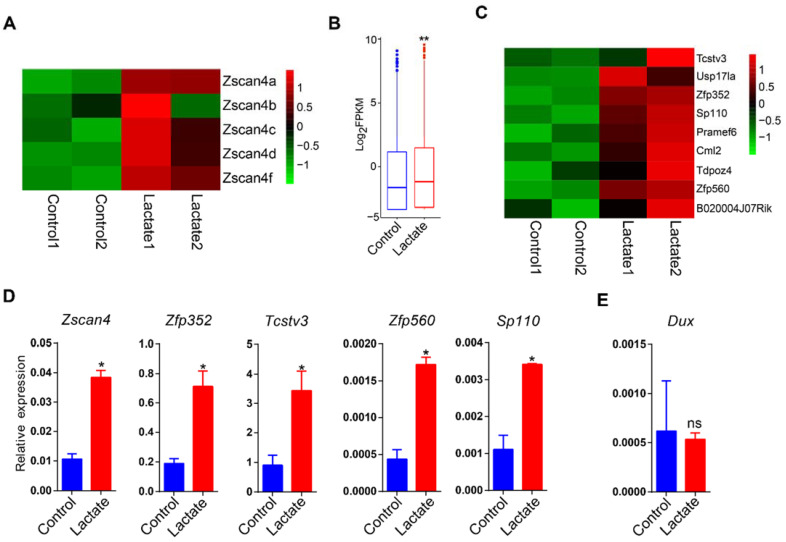
Lactate supplementation promoted expression of ZGA genes in mouse ESCs. (**A**) A heatmap of expression of *Zscan4* family genes. (**B**) Global expression comparison of upregulated genes in MERVL^+^Zscan4^+^ ESCs between control and lactate-supplemented mouse ESCs. The Wilcoxon rank sum test was used to calculate *p*-values. ** *p* < 0.01. (**C**) Heatmap of representative upregulated ZGA genes. qRT-PCR analysis of ZGA genes (**D**) and *Dux* gene (**E**) in control and lactate-supplemented mouse ESCs. Data are presented as means ± SD (*n* = 3); * *p* < 0.05; ns, *p* >0.05. Student’s *t*-test was used to calculate *p* values. *Actb* was used as the internal control.

**Figure 4 cells-11-00548-f004:**
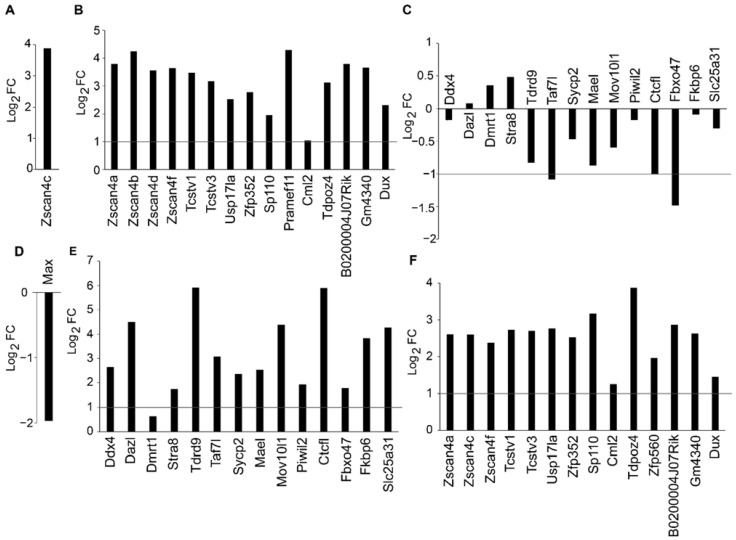
Sequential activation of germline genes and ZGA genes. (**A**–**C**) Analysis of public RNA-seq data (GSE120998) for *Zscan4c* overexpression (**A**); ZGA gene expression change (**B**); germline gene expression change (**C**). (**D**–**F**) Analysis of public microarray data (GSE45181) for *Max* downregulation (**D**); germline gene expression change (**E**); ZGA gene expression change (**F**).

**Figure 5 cells-11-00548-f005:**
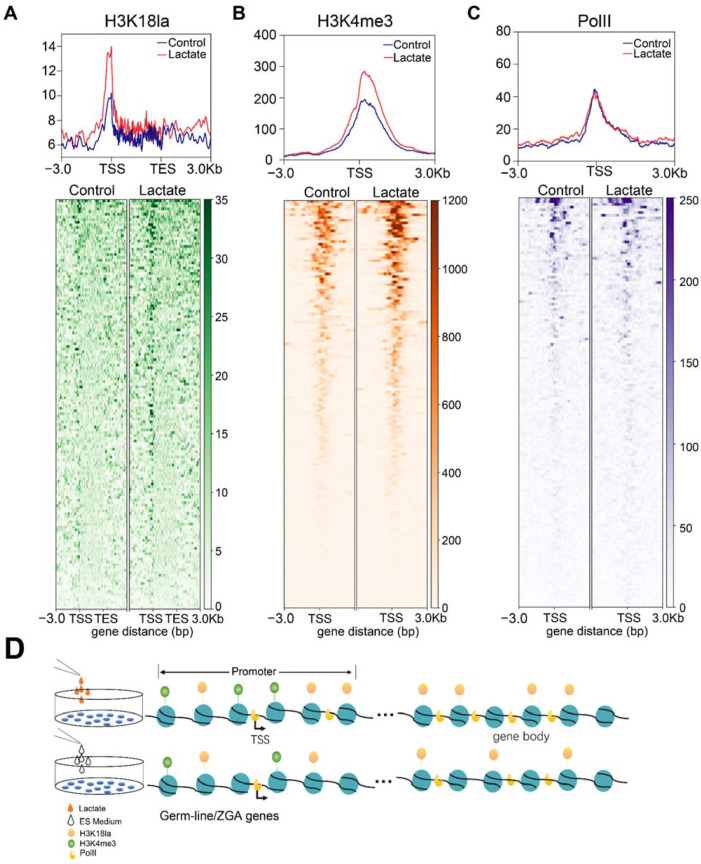
Lactate supplementation facilitated transcriptional elongation through enhanced histone lactylation. (**A**–**C**) Read density plot of H3K18la (**A**), H3K4me3 (**B**), and RNA PolII (**C**) signals at gene bodies and promoters of upregulated genes (**upper panel**), and corresponding heatmap at gene bodies and promoters (3 kb flanking TSSs) of upregulated genes (**lower panel**). (**D**) Schematic diagram of the mechanism by which lactate promotes expression of germline/ZGA genes.

## Data Availability

RNA-seq and ChIP-seq data generated during this study were deposited to GEO database at December 2021: GSE191252, GSE192358.

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
