# Peer review of "Lactate Activates Germline and Cleavage Embryo Genes in Mouse Embryonic Stem Cells"

_cells, 2022, doi:10.3390/cells11030548_

Round 1
Reviewer 1 Report
cells-1572826 Jan 2022 review comments
Overview:
Here, the authors tested the effects of lactate on embryonic stem cell pluripotency and gene expression. Lactate supplementation seems to shift ESCs towards the germ line through epigenomic and subsequent transcriptomic changes.
Major:
The effects of lactate in culture media can vary depending upon the concentration, whether there are low (sub-physiological) or high (supra-physiological) levels. Why was 50mM lactate chosen? Do you have empirical validation that this is an appropriate amount? I am concerned that adding too much, or too little lactate, may have off-target effects and not replicate what occurs in vivo.
In lines 122-129, the authors correlate higher H3K18la levels with “transcriptional activation”. However, just because a gene is highly expressed does not necessarily mean it is highly activated in this system (by lactate supplementation) – it may already be highly expressed and so it’s transcriptional activity is unchanged. Please clarify whether you are correlating high H3K18la with high expression, or high transcriptional activation.
The authors assert that lactate supplementation facilitates differentiation of ESCs towards germ line cells (lines 151-152), based on RNA-seq and qPCR gene expression changes. This is a good observation but to really bolster your assertion, functional evidence of the shift towards germ line state would be beneficial.
It appears that RNA-seq was performed on n=2 control or lactate samples. Is this sufficient for statistical analysis and interpretation?
The data supporting Figure 5’s conclusions are lacking strength. Pol II is not enriched at the TSS of up-regulated genes (although it is in the gene body), which is strange, but not completely unusual. Further testing may clarify these results, possibly through chromatin accessibility assays to assess how open chromatin at the TSS of up-regulated genes is.
In line 211, it is stated that one in sex couples worldwide are infertile. This seems to be quite high, more than I would expect. If it is true, could you please provide citations?
Minor:
For RNA-seq methods, could you please clarify how adjusted P value is calculated in DESeq2? Is it based on False Discovery Rate?
For statistical analysis, can you justify why student’s t-tests were used? Were normality tests performed? Were all data a normal distribution, warranting a parametric test?
Avoid using terms like “and so on” or “etc”
What tool was used to perform gene ontology analysis? Please state in Methods, and if possible, provide GO term IDs in supplements.
Please provide more details on the public RNA-seq and microarray datasets.
In line with my suggestion about improving Figure 5’s supporting data, please check your schematic model to ensure that it aligns with your conclusions. Eg. does the TSS encompass all three histones or just the one directly above the label?
Reviewer 2 Report
In their paper entitled “Lactate Activates Germline and Cleavage Embryo Genes in Mouse Embryonic Stem Cells”, the Authors report that lactate supplementation activates germline genes in mouse embryonic stem cells (ESCs). Moreover, lactate induces up-regulation of cleavage embryo genes probably through lactylated H3K18 accumulation on germline and cleavage embryo genes, thus promoting transcription.
The paper is of great interest, and both methods used and results are clearly described. I only suggest a couple of very minor modifications:
- Introduction: the last paragraph (lines 49-58), that concerns lactate, could be a bit improved by briefly referring to the already known functions of lactate, and, possibly, to the metabolic mechanisms that have been proposed as possible biochemical links between lactate and lactylation (i.e. lactyl-CoA and enzyme-dependent reaction, and non-enzymatic, lactyl-glutathione-dependent reaction. It has been also proposed that p300 is involved. Please, note that such information is, by the way, reported in their ref. 24: Chen et al., Front Immunol. 2021; 12:688910);
- Figures 2 and 4: letters and numbers are too small and difficult to read.
Round 2
Reviewer 1 Report
Even though I would have liked to see further functional validations of their conclusions, I recommend this manuscript for acceptance given that the authors have addressed most of my concerns.